# Marker-Free, Molecule Sensitive Mapping of Disturbed Falling Fluid Films Using Raman Imaging

**DOI:** 10.3390/s22114086

**Published:** 2022-05-27

**Authors:** Marcel Nachtmann, Daniel Feger, Sebastian Sold, Felix Wühler, Stephan Scholl, Matthias Rädle

**Affiliations:** 1Center for Mass Spectrometery and Optical Spectroscopy, Hochschule Mannheim University of Applied Sciences, 68163 Mannheim, Germany; d.feger@hs-mannheim.de (D.F.); s.sold@hs-mannheim.de (S.S.); f.wuehler@hs-mannheim.de (F.W.); m.raedle@hs-mannheim.de (M.R.); 2Institute for Chemical and Thermal Process Engineering, Technische Universität Braunschweig, 38106 Braunschweig, Germany; s.scholl@tu-braunschweig.de

**Keywords:** falling film, Raman spectroscopy, non-contact, flow characteristics, marker-free, molecule sensitive

## Abstract

Technical liquid flow films are the basic arrangement for gas fluid transitions of all kinds and are the basis of many chemical processes, such as columns, evaporators, dryers, and different other kinds of fluid/fluid separation units. This publication presents a new method for molecule sensitive, non-contact, and marker-free localized concentration mapping in vertical falling films. Using Raman spectroscopy, no label or marker is needed for the detection of the local composition in liquid mixtures. In the presented cases, the film mapping of sodium sulfate in water on a plain surface as well as an added artificial streaming disruptor with the shape of a small pyramid is scanned in three dimensions. The results show, as a prove of concept, a clear detectable spectroscopic difference between air, back plate, and sodium sulfate for every local point in all three dimensions. In conclusion, contactless Raman scanning on falling films for liquid mapping is realizable without any mechanical film interaction caused by the measuring probe. Surface gloss or optical reflections from a metallic back plate are suppressed by using only inelastic light scattering and the mathematical removal of background noise.

## 1. Introduction

In the chemical industry, plant design includes a variety of simulation efforts to ensure proper plant behavior. Especially in multi-component thermal separation units, several assumptions concerning the fluid product behavior have to be made, as parameters are derived from few known compounds. A regular occurring situation is that a new developed product with unknown fluid parameters is synthesized in a mixture of fluids. During the purification process, the product is separated from the rest of the mixture. This is particularly true regarding separation equipment for production processes, such as falling film evaporators, but also in, for example, drying towers and many other aggregates in the process engineering industry. The goal of the work presented here is to provide a new control method to experimentally determine the flow behavior or even reaction tracing, and for this to present comparison data to validate the upcoming numerical simulations. For a complete description of the falling films and to provide all needed information for a purposeful development of technical aggregates, contact-free, local, molecular sensitive, marker-free concentration profiles, and other fluid parameters such as local film thickness are essential. Raman spectroscopy meets all the above requirements, but was not applied for scanning measurements in falling films. The reason, perhaps, can be found in the necessary long integration times and therefore, extremely slow scanning velocity. Another reason might be the necessity of the plant being in a steady state. Any changes during measurements interfere with the acquired data. To solve this issue, a designing probe heads for long distances and broad aperture, resulting in an improved photon counting efficiency. On the detection side, for this work, a process CCD Raman spectrometer is used. Process spectrometers lack resolution compared to research Raman spectrometers, but allow fiberglass-connected, flexible usable probes, and have the ability to detect all wavelengths simultaneously. They provide an improved robustness to external influences, especially mechanical vibrations. To realize automatic flow mapping, the probe has been attached to a motorized linear stage, the excitation source has been altered, and other adjustments have been made. This publication presents measurements of the liquid film profiles along a cross-section on a plain surface falling film, and with a pyramid-shaped disturbance. The following sections give a short overview of the state of the art and provide some theoretical background.

### State of the Art

Three-dimensional imaging is a relevant topic in all kinds of research and many parts of the industry. Many applications are presented in the open literature. Without going beyond the boundaries of the presented work, the focus is on methods of Raman imaging in particular, but also methods for marker-free detection of flow behavior. This section presents the state of the art directly connected by measurement techniques or applications.

Zhao et al. used halftone spatial frequency domain imaging to detect optical properties on turbid media. Scattering and absorption information were obtained and interpreted. The focus was on biological samples. In contrast to the method presented, Zhao’s method enabled imaging of a tissue. Chemical properties were not measured and can only be determined indirectly using the reduced scattering as an indicator [1].

Dhong et al. measured fluids, bubbles, and particles in microchannels using metallic nano-islands on graphene. The presence of bubbles and particles are detected by the flow behavior at the graphene sensor. By increasing the number of sensors, it is possible to measure the flow behavior. There is no molecule detection possible [2].

Regarding falling films, Mohamed characterized the flow behavior in horizontal rotating tubes. Compared to the presented approach, both methods require stationary films for detection. The film thickness was calculated using images made by a camera. There is no spectral information or 3D information possible. The goal of this study was to measure the influence of rotation speed on the Reynolds number and the reduction of a dimensionless wavelength [3].

Medina et al. presented an approach for a local concentration and viscosity measurement in microchannels using NIR spectroscopy. In this approach, the investigation of flow behavior is also possible, due to the availability of the layer thickness. The liquids are excited using different LEDs as the light source. The reflectance and absorbance are measured with a NIR camera. The approach offers a fast 2D image of the liquids present. Due to the different absorbance values, a statement about the molecules present is possible [4].

In the field of Raman imaging, there are only few publications, because Raman imaging is, compared to laboratory use, not commonly used in the process industry. Raman spectroscopy itself is also a niche technology in the process industry and is only used in very specific tasks.

Greszik et al. measured film thickness using laser-induced fluorescence (LIF) and Raman imaging. LIF measurements require an organic tracer added to water, as pure water has no fluorescence. For Raman scattering, no tracer is required. Measuring water, both systems create a 2D image. For film thickness measurements, a static film was applied to quartz plates. Although the measurement technique (Raman imaging) is identical, the approach differs in various ways. At first, only static liquids are measured, while in this work, sodium sulfate in water falls down a falling film surface. The optical setup contains an ICCD camera for LIF and Raman imaging. Compared to the usage of a Raman spectrometer, the measurements are faster, but lack detailed spectral information [5].

Sovany et al. used Raman spectroscopy for film thickness regarding pharmaceutical preparations. The goal was to determine the thickness of polymer coating on pellets. In the research presented, a fiber probe was used, whose measuring spot was larger than one pellet [6].

Wiliams et al. presented a possibility for Raman imaging in the process environment using a Raman microscope. The applications presented ranged from thin surface films, as well as polymer laminates, up to material contamination in plastics. The researchers used the direct imaging capability of the Raman microscope. In this publication, we used a Raman probe combined with a motorized linear stage for point-by-point measurements. Compared to the direct imaging, not only thin layers or surface measurements are possible. Another limitation of the system is the short working distance of the microscope used [7].

Stewart et al. released a review on Raman imaging in 2012. The publication sums up many different 2D approaches. Those approaches focus on quality control in biomedical and pharmaceutical industries as well as threat-detection. Stewart et al. also described the point-by-point approach used in this publication but only in two dimensions. The chemical process industry or measurements during chemical operation are not described. One reason might be, that point-by-point imaging measurements require long measuring times, thus need a steady state of the sample measured. Another downside might be the possible damage to the sample. In the presented, due to the flowing film, no damage is possible, however, the steady state could be reached [8].

Regarding the current state of the art, no method for a marker-free, non-contact, and molecule selective method for 3D imaging on falling films could be found. In the presented work, reference is made to 2D slides, but unlike the state of the art, these slides are measured from the air through the film to the surface of the falling film, forming a cross-section at a specific position. Other approaches offer surface images but no depth information. Unlike approaches measuring common 2D images, a real 3D image is possible.

## 2. Materials and Method

### 2.1. Raman Effect

For a better understanding of the presented work, the basic principle of the Raman effect is briefly explained. Raman imaging as well as Raman spectroscopy rely on this effect. The Raman effect describes inelastic scattering of electromagnetic radiation, subclassified into Stokes and anti-Stokes scattering. The Raman effect itself was discovered by C.V. Raman [9,10]. The presented approach utilizes Stokes scattering, as it is the dominant Raman effect. A laser as an excitation source can raise a molecule from the ground state to a virtual energy level. To achieve disexcitation, the molecule emits a photon and returns to a vibrational excited state, but not the ground state. A molecule’s disexcitation to the ground state is an elastic scattering effect, the so-called Rayleigh scattering. Stokes scattering is the emitted radiation caused by the disexcitation to the vibrational state. The emitted radiation has a higher wavelength and thus, a lower frequency than the excitation source. For every functional group or covalent bond, the wavelength shift (excitation to scattering) is characteristic and relative to the excitation wavelength. This provides information about the molecules detected. The intensity of the Raman scattering bonds is directly proportional to the intensity of the excitation laser and its excitation power. Other dependencies also apply [11,12,13,14,15,16]. The mathematical correlation of the intensity with the excitation frequency *ν*, the excitation power I_0_, the number of scattering molecules n, and polarizability of the molecules $\frac{\delta_{a}}{\delta_{q}}$ is given by the following formula [16]:(1)$$I\;\propto {\;\nu }^{4}\;\cdot {\;I}_{0}\;\cdot \;n\;\cdot \frac{\delta_{a}}{\delta_{q}}$$

### 2.2. Falling Films

Falling films are trickle films with a slope of 90 degrees. An example for a falling film is shown in Figure 1. While trickle films with different slopes are common in process development, with a slope of 90 degrees, these trickle films fill a niche for thermal unstable products. The thin film offers a high heat exchanger surface to liquid volume ratio. It is often used in falling film evaporators, however, also mixing falling films are possible. The most important characteristic number is the minimum wetting density $\frac{\dot{V}}{B}$, with the volume flow rate $\dot{V}$ and the wetting density *B* [17,18,19,20].

The kinematic viscosity *ν*, the acceleration of gravity *g*, and the film thickness *δ*, define the minimum wetting density in the following simplified correlation [20]:(2)$$\frac{\dot{V}}{B}=\frac{g*\delta^{3}}{3\nu_{f}}$$

In literature, minimum wetting density under ideal conditions should be between 0.5 $\frac{m^{3}}{mh}$ and 1.5 $\frac{m^{3}}{mh}$ [20].

Reynolds number for a specific liquid is calculated using V˙B, minimum wetting density, *B*, characteristic length perpendicular to the flow direction, and *ν*, kinematic viscosity [20]:(3)$$Re=\frac{\frac{\dot{V}}{B}}{\nu }$$

Film thickness is a correlation of kinematic viscosity and Reynolds number as well as acceleration of gravity [20]:(4)$$\delta ={\left(\frac{\nu^{3}}{g}\right)}^{\frac{1}{3}}*Re^{\frac{1}{3}}$$

The minimum wetting density is specific to a given component system, heat transfer intensity and surface characteristics, and is determined for the individual plant. As minimum wetting density, Reynolds number and film thickness are interdependent, and the experimental determination of at least one characteristic number is necessary [20].

### 2.3. Experimental Plant

The experimental plant contains a falling film, connected to an overflow reservoir. This reservoir consists of two separated chambers. A rotary vane pump (AFT GmbH & Co KG/Rota, Germany) feeds the reservoir using one pump for each chamber. Frequency converters (Siemens AG/Munich, Germany) allow fine tuning the flow rate. After filling the reservoir, the liquid starts flowing down the vertical surface. In this particular study, filling both chambers with an identical sodium sulfate solution to observe flow behavior is sufficient. By using separated chambers, the falling film is ready for mixtures or chemical reaction measurements. The falling film itself is made of stainless steel, and the overflow reservoir is made of aluminum. A schematic diagram experimental plant is shown in Figure 1.

The pump conveys the fluid into the overflow container (1). As soon as it is filled to the maximum, the fluid runs down the vertical wall of the falling film (2). In the center of this vertical wall is an aluminum pyramid (3), which acts as a flow obstacle. The Raman probe (4) is used to measure the liquid behavior around this pyramid and is connected to a motorized linear stage. Therefore, the Raman probe is connected to a laser source (5) and a spectrometer (6).

The pyramid is shown in Figure 2 and is made out of aluminum. Many different disturbances are possible. The individual plant component dimensions, relevant for the presented measurements, are listed in Table 1.

A motorized linear stage, with a Raman probe attached, allows optical measurements. To ensure stable and reproducible measurements, a direct connection to the falling film is favorable for the motorized linear stage. Regarding the motorized linear stage, the minimum step size is 10 µm and reproduction accuracy is 5 µm. A LabVIEW (National Instruments Corp/Austin, TX, USA) program connects and synchronizes the motorized linear stage with the Raman spectrometer. The spectroscopical setup consists of a Multispec Raman spectrometer (tec5 AG/Steinbach, Germany), and an external single-mode laser source (Integrated Optics/Vilnius, Lithuania). The Raman shift was calculated using a reference measurement with cyclohexane. Re-engineering the Raman co-axial probe was conducted by using one-inch optics for increased numerical aperture to increase signal strength as well as to reduce depth of focus. The optical components and their parameters are listed in Table 2.

### 2.4. Design of Experiments

Sodium sulfate emits a strong Raman signal and offers various advantages as a reference substance. It is non-toxic, soluble in water, fluorescence-free, and has an isolated peak around 950 cm^−1^ Raman shift. Its Raman spectrum is shown in Figure 3. The spectrum was acquired with the same equipment as the presented measurements.

Measurements focus on three horizontal slides, providing a transverse flow section each: one slide above, below, and in the center of the pyramid. Regarding the presented measurements, a point-to-point measurement approach was favored, due to its ability for real 3D-scanning. The respective measuring ranges and step sizes between the measuring points are listed in Table 3. Due to the increased height, the pyramid slide required a larger measurement window compared to the other slides. Limiting the traverse path on slide 1 and 3 in the Z-axis, and therefore scaling down the measurement windows, allows time-saving in measurement cutting only areas with no sodium sulfate film present.

For every slide, the liquid and optical parameters are identical. For a better overview, the parameters are listed in Table 4. The liquid parameters are plant specific and were experimentally determent. The combined flow rate of 100 L/h ensures a stable film with as little turbulence as possible. Sodium sulfate can be measured in a concentration range from 0 mol/L up to its saturation in water, ca. 1.2 mol/L. The concentration was set to 1 mol/L for a sufficient Raman signal but also to ensure proper solubility without any unsolved particles. The intensity and peak position for data analysis was calibrated prior to the measurements at process temperature. Regarding the optical parameters, integration time is always a trade-off between measurement time and signal strength. With 5 s integration time for an individual spectrum, an optimum for the measurements presented could be found. For noise reduction, the spectra were accumulated two times.

### 2.5. Data Analysis

The data analysis utilizes an in-house developed Python 3 tool (Python Software Foundation, Wilmington, DE, USA). This generates hyperspectral datacubes from the point-wise acquired spectroscopic data by assigning a corresponding image pixel to each spectrum. In order to obtain a better evaluation of the desired Raman peaks, a pre-processing of the Raman spectra was subsequently performed, using only established methods of several researchers [21,22,23,24]. Limiting the spectral range from 750 cm^−1^ to 1250 cm^−1^ eliminates both the zero values due to software—which would cause errors in the further processing steps and the spectral ranges without relevant information. This reduces the number of spectral values from the original 3200 to 500. As a positive side effect, the performance, due to the reduced amount of data, is increased. Subsequently, a spectrum despeckling algorithm removes the cosmic rays [25]. These spikes turned up randomly when measured with the Raman spectrometer due to high energy particles hitting the CCD detector. In a next step, an offset correction was performed for each spectrum to the level of the lowest intensity value. To compensate for the background offset, an asymmetric least square smoothing achieves a baseline correction. This was published by He et al. [26] in 2014. The applied algorithm runs over 10 iteration levels with the values *p* 0.1 as well as lambda 1,000,000. In the last step of the pre-processing, an attempt was made to reduce the noise of the Raman spectra, using a Savitzky–Golay filter [27] with a 3rd order polynomial fit and a window size of 11. Visualizing the desired wavelengths included in the data treated, the counts from minimum to maximum are assigned to a color scheme and converted to 0% to 100%, as shown in the legend of Figure 4. The developed Phyton 3 tool provides all pre-processing needed as well as the image generation in one operation interface.

## 3. Results and Discussion

After the measurements, the raw spectra received contained background noise as well as some elastic scattering passing the edge filter. The python data analysis program removed the cosmic spikes as well as the background noise. Sodium sulfate has a well isolated peak at a Raman shift of 950 cm^−1^. Water has a broad peak above 3200 cm^−1^. Due to detection limits, only the sodium sulfate peak was accessible. Heatmaps with pseudo coloring provided great visibility and information on the measurements presented. To achieve comparability, a full set of all three slides was recorded at a time. Each picture contained 1681 measurement points and took about 5 h to complete. To ensure laminar flow, but also to spot any turbulence caused by imperfections on the plain surface, the film was visually inspected before every measurement cycle. Comparing the individual measurement slides to the corresponding measurement in other cycles, a statement about reaching steady state was possible. Upon reaching steady state, the measurements from different cycles were comparable. Figure 4 shows an exemplary measurement processed as stated.

The minimum wetting density was determined experimentally and the pumps were both set to 50 L/h to ensure a stable film. One plain surface measurement was located upstream of the pyramid-shaped disturbance, and the other downstream the disturbance. Comparing the 2D slides, the influence of the pyramid-shaped disturbance was clearly distinguishable. During laminar flow, the sodium sulfate film stuck to the surface of the falling film. There was no liquid splashing caused by the pyramid. The pyramid itself only affected the film during the disturbance and shortly after it. Moving down the film, the influence diminished.

The results of the first slide are displayed in Figure 4. Sodium sulfate (middle section of the diagram, marked with black bars), air (left section of the diagram), and the surface (right section of the diagram) form sharp borders between their respective locations in the measurement slides. As shown in Figure 4, a plain film was detected. Prior to the disturbance (Figure 4), the liquid film was about 0.8 mm, which is 0.36 mm thicker than theoretically calculated for pure water. This might be due to imperfections on the surface and non-uniformity in the liquid feed. Rotary vane pumps tended to a light pulsation increasing with low rotation speeds. The pulsation formed light waves. Another influence might be the usage of tap water instead of distilled water. Last but not least, the film boundaries can be blurred in a limited range due to the depth of focus. The depth of focus indicates the length of the focal point. Using the custom-engineered probe, the depth of focus was 100 µm.

Following plain surface measurements, a 2D sheet was taken on top of the pyramid-shaped disturbance (see Figure 5). The disturbance was spatial after the plain surface 2D sheet. Next to the disturbance, the film dammed up for about 0.2 mm; on top of the pyramid, the film was as thick as the undisturbed film, and on the rising sides, the film was about 0.2 mm to 0.3 mm thinner than the undisturbed film. The damming next to the pyramid was due to liquid following the path of least resistance, as well as the thinner film on the rising sides. As a triangle shape leading to the top of the pyramid, the top of the pyramid also had a lower liquid resistance than the pyramid sides. This was due to the pyramid’s orientation. Different orientations produced different flow characteristics.

Last but not least, another plain surface measurement was conducted to distinguish if the pyramid impacts the film behavior falling further down the falling film surface. In Figure 6, measurements downstream of the pyramid are shown. The film was about 0.8 mm thick. In the middle of the measured film, below the pyramid, the film thickened for about 0.05 mm to 0.1 mm. The depth of focus caused blurred boundaries again for around 0.1 mm.

During data analysis, the background can be completely suppressed. The raw spectra contained a high amount of background noise, especially due to the stainless-steel surface. The cleaned spectra showed no background noise.

## 4. Conclusions

In conclusion, measurements on falling films using Raman spectroscopy were possible with a good signal-to-noise ratio. For a sodium sulfate solution, the wavenumber of 950 cm^−1^ was, as expected, best suited, due to the maximum intensity of the signal. The measured film thicknesses were in the range of 0.8 mm with small deviations directly around the pyramid. Theoretically, a film thickness of 0.44 mm was calculated. When using two or more different liquid streams, mixtures were also spectrally available, as Raman spectroscopy is molecule selective.

The measurement set-up is not limited to sodium sulfate. A wide range of liquids or substances in solution are accessible. As restrictions, the relevant substances must be Raman active and not show any fluorescence in the wavelength region of the excitation laser. Fluorescence is a severe issue when using Raman spectroscopy, as it is the dominant effect and overshadows the Raman signal. In this publication, only liquid 2D slides are presented. In the future, the goal is to measure three-dimensional slides. With 3D slides available, a better understanding of flow behavior around distinct disturbances can be achieved. There are only a few limitations regarding the shapes of the disturbances, however, optical availability of the liquid stream is mandatory. Depending on the substances and the depth of field needed, encapsulated plants with glass windows on relevant parts of the falling film are available for measurements, as the working distance is variable.

Measurement times, and therefore, signal-to-noise ratio, as well as laser power provided potential improvements. Thus, Raman scattering is directly proportional to the optical power of the excitation laser, and a brighter laser source reduces measurement times. To obtain a small spot and a small depth of focus, as well as to uncouple the laser from the motorized linear stages’ movement, a single-mode fiber-coupled laser is the favored excitation source. Compared to a multi-mode laser, the single-mode lasers are power restricted, as only one wave passes through the glass fiber. To remain in single-mode and increase laser brightness, a free space laser source is required. This can lead to stability problems. For many applications, a higher energy intake is possible. Hence, falling films are commonly used for thermally unstable substances, as the higher energy intake might alter or even damage the chemicals irreversibly. There is no limitation to falling films of the presented approach, and the measurement system is available to all set-ups with optical access on relevant areas. The re-engineered Raman probe is currently optimized for a trade-off between small depth of focus, small focal point, and signal strength. Depending on the application, signal strength, focal point or depth of focus is favored, and the probe can be re-engineered to meet the requirements. Another influence on measurement times is the speed of the motorized linear stage. Regarding one measuring point, the moving times might seem insignificant, but for a full measurement cycle, even small numbers add up.

## Figures and Tables

**Figure 1 sensors-22-04086-f001:**
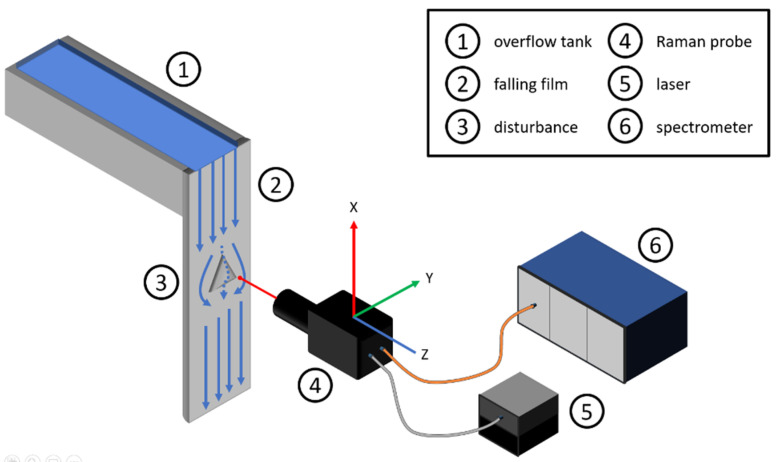
Schematic diagram of the experimental plant containing the overflow tank (1), the falling film itself (2), the disturbance as well as the Raman probe (4), the laser as excitation source (5), and the Raman spectrometer.

**Figure 2 sensors-22-04086-f002:**
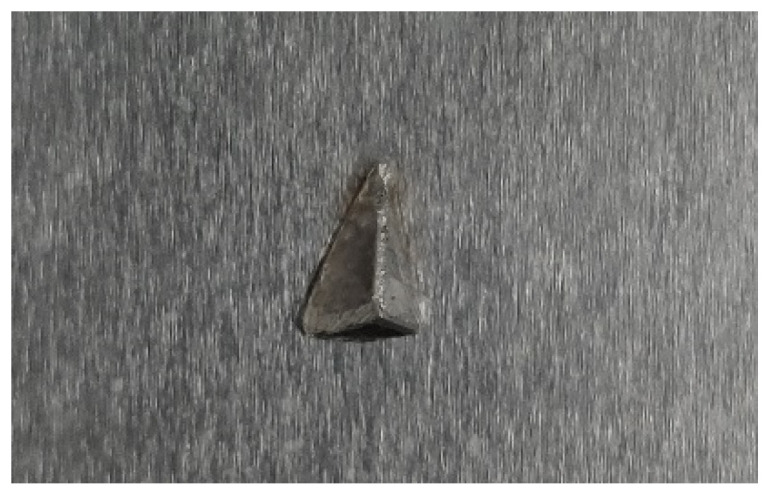
Pyramid attached to the falling film surface as an example of a film disturbance.

**Figure 3 sensors-22-04086-f003:**
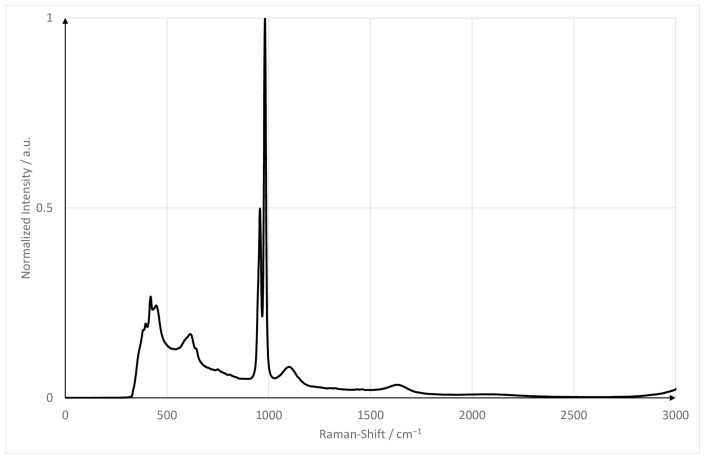
Raman spectrum of sodium sulfate hydrate.

**Figure 4 sensors-22-04086-f004:**
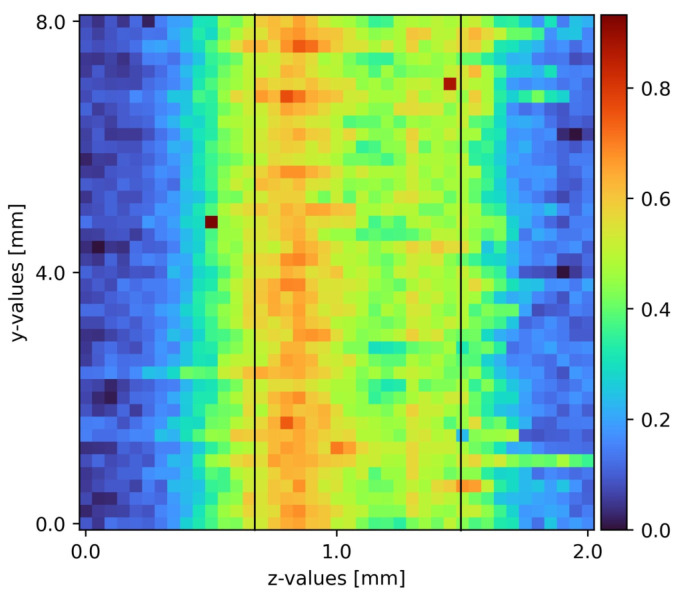
Sodium sulfate film on a plain surface falling film spatial before the disturbance. The black bars indicate the boundaries of the sodium sulfate film with air left on the left and the surface at the right.

**Figure 5 sensors-22-04086-f005:**
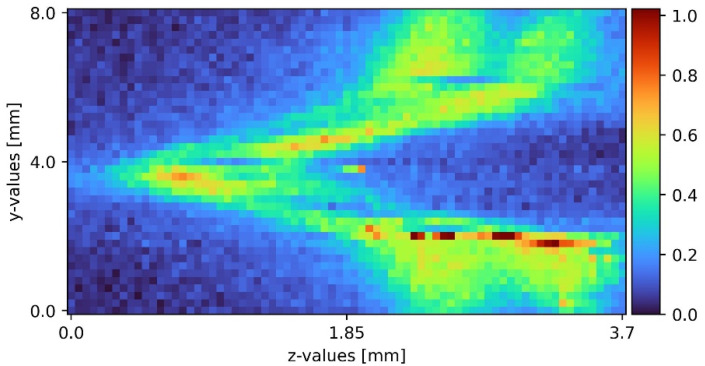
Sodium sulfate film on top of a pyramid-shaped disturbance.

**Figure 6 sensors-22-04086-f006:**
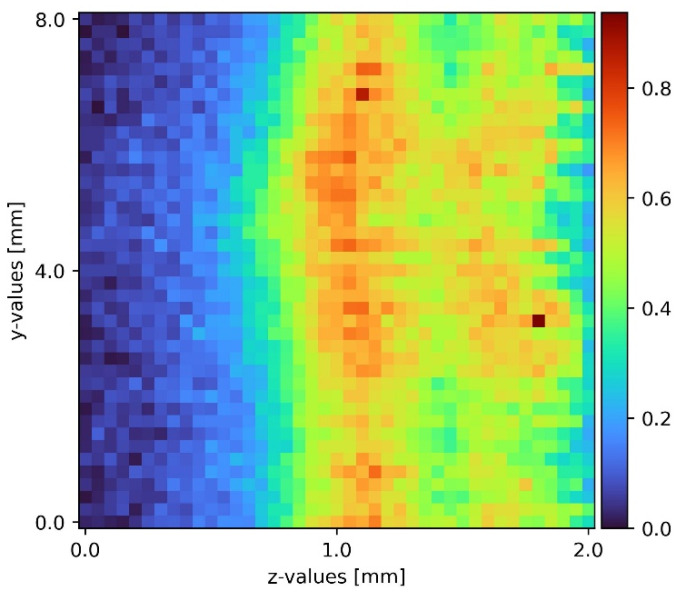
Sodium sulfate film on a plain surface falling film; measurements are located after the disturbance.

**Table 1 sensors-22-04086-t001:** Experimental plant dimensions.

Plant Component	Parameter	Value
Falling film surface	x	600 mm
	y	100 mm
	Minimum wetting density	$1\;\frac{m^{3}}{mh}$
	Calculated film thickness	0.44 mm
Pyramid	x	6.5 mm
	y	4.1 mm
	z	2 mm
Pump	Type	Rotary vane
	Maximum flow rate	150 L/h
	Maximum power outlet	550 W

**Table 2 sensors-22-04086-t002:** Optical components.

Optical Component	Parameter	Value
Raman spectrometer	Wavelength	785 nm
	Optical range	320–3200 cm^−1^
	Optical resolution	5 cm^−1^
Laser source	Wavelength	785 nm
	Power	130 mW
	Optical coupling	Single-mode coupling
Raman probe	Focal point	<40 µm
	Depth of focus	75 µm
	Working distance	16 mm
	Type	Co-axial

**Table 3 sensors-22-04086-t003:** Traversing parameters for the different 2D slides.

Slide	Axis	Distance	Step Size
1 (top)	X	0	0
	Y	8 mm	0.2 mm
	Z	2 mm	0.05 mm
2 (pyramid)	X	0	0
	Y	8 mm	0.2 mm
	Z	3.8 mm	0.05 mm
3 (bottom)	X	0	0
	Y	8 mm	0.2 mm
	Z	2 mm	0.05 mm

**Table 4 sensors-22-04086-t004:** Plant and optical parameters.

System	Parameter	Value
Liquid	Sodium sulfate	1 mol/L
	Water	50 L
	Temperature	20 °C
Optics	flow rate	100 L/h
	Integration time	5 s
	Accumulation	2

## Data Availability

Not applicable.

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
