# Peer review of "Marker-Free, Molecule Sensitive Mapping of Disturbed Falling Fluid Films Using Raman Imaging"

_sensors, 2022, doi:10.3390/s22114086_

Round 1

Reviewer 1 Report

This paper presents an experiment to elaborate promising application of Raman spectroscopy, as an effective detection technology, on molecule sensitive, non-contact and marker-free localized concentration mapping in vertical falling films which does not need to label or mark when applying in composition of liquid mixtures. It looks that this interesting work provides a new tool or method for measuring and observing liquid flow films. There have some puzzles in its content which I cannot understand clearly.

The main comments are as follows:

1) In the test, authors list optical components and parameter values. Whether does different optical conditions affect data accuracy and results, for example, step size, reproduction accuracy, etc.?

2) Can you give the applicable concentration scope of sodium sulfate solution if others want to repeat this measure? And, what other types of chemical solutions do you think can be tested similarly.

3) The liquid temperature is 20 °C. I guess the test should be done at atmospheric pressure. Therefore, whether will the thermodynamic conditions affect your conclusion, such as high pressure or low temperature?

4) The influence of plant morphologicalcharacteristics. If the dimensions of film surface and pyramid change, whether is the signal-to noise ratio well controlled?

Reviewer 2 Report

See attached file.

Round 2

Reviewer 1 Report

These responses have completely explained my doubts on the experimental design, its results and final conclusion in this manuscript. Thanks to authors for their excellent work. Now I agree to accept it in Sensors.

Reviewer 2 Report

I recommend manuscript for publication.